# Chemical Composition of Green Pea (*Pisum sativum* L.) Pods Extracts and Their Potential Exploitation as Ingredients in Nutraceutical Formulations

**DOI:** 10.3390/antiox11010105

**Published:** 2021-12-31

**Authors:** Luigi Castaldo, Luana Izzo, Anna Gaspari, Sonia Lombardi, Yelko Rodríguez-Carrasco, Alfonso Narváez, Michela Grosso, Alberto Ritieni

**Affiliations:** 1Department of Pharmacy, University of Naples “Federico II”, 49 Domenico Montesano Street, 80131 Naples, Italy; annagaspari@virgilio.it (A.G.); sonia.lombardi@unina.it (S.L.); alfonso.narvaez@unina.it (A.N.); 2Laboratory of Food Chemistry and Toxicology, Faculty of Pharmacy, University of Valencia, Av. Vicent Andrés Estellés s/n, 46100 Burjassot, Spain; yelko.rodriguez@uv.es; 3Department of Molecular Medicine and Medical Biotechnology, School of Medicine, University of Naples “Federico II”, 5 Sergio Pansini Street, 80131 Naples, Italy; michela.grosso@unina.it; 4Department of Pharmacy, University of Naples Federico II, Via Domenico Montesano 49, 80141 Napoli, Italy; alberto.ritieni@unina.it

**Keywords:** bioactive compounds, bioaccessibility, food waste valorization, pea, polyphenols, nutraceutical

## Abstract

Agro-industrial wastes contain a large number of important active compounds which can justify their use as innovative ingredients in nutraceutical products. This study aimed to provide a complete analysis of active molecules, namely (poly)phenols in pea pods water-based extracts, through a UHPLC-Q-Orbitrap HRMS methodology. Data showed that 5-caffeoylquinic acid, epicatechin, and hesperidin were the most relevant (poly)phenols found in the assayed extracts, with a mean value of 59.87, 29.46, and 19.94 mg/100 g, respectively. Furthermore, changes in antioxidant capacity and bioaccessibility of total phenolic compounds (TPC) after the simulated gastrointestinal (GI) process were performed using spectrophotometric assays (FRAP, DPPH, ABTS, and TPC by Folin-Ciocalteu). The acid-resistant capsules (ARC) and the non-acid resistant capsules (NARC) containing the pea pod extract underwent simulated GI digestion. The results suggested that the ARC formulations were able to preserve the active compounds along the simulated GI process, highlighting a higher TPC value and antioxidant capacity than the NARC formulations and the not-encapsulated extracts. Hence, the pea pods water-based extracts could be utilized as a potential alternative source of active compounds, and the use of ARC could represent a suitable nutraceutical formulation to vehiculate the active compounds, protecting the chemical and bioactive properties of (poly)phenols.

## 1. Introduction

Pea (*Pisum sativum*) is a cool-season plant belonging to the Fabaceae family widely cultivated all over the world and commonly considered native to Southern Europe [1]. Pea seeds have been long consumed due to their nutritional value, which includes a high content of proteins, starch, fibers, minerals, and vitamins [2]. Nevertheless, pea seeds contain antinutritional factors such as phytic acid, which is related to impaired mineral absorption. The world yearly production of green peas reached about 21.22 million tons in 2018. Consequentially, very large amounts of agro-waste are unavoidably generated in the pea industry. Currently, the exploitation of agro-wastes and by-products has aroused considerable interest since it represents an eco-friendly and innovative way to reduce the environmental impact and promote Sustainable Development Goals [3] being in agreement with the United Nations Member States Agenda. Generally, procedures that require a simple water-based extraction are considered as the greenest approaches to reduce the environmental impact and to recover active molecules from agro-industrial wastes [4,5,6]. In fact, these vegetal residues contain an important amount of health-promoting compounds that may be exploited effectively as ingredients in the formulation of dietary supplements, nutraceutical products, or fortified foods [7,8,9].

The main residue emanating from the pea industry is represented by the pods, which make up about 30–67% of the total weight of the whole pod [7]. Previous experimental studies have reported that pea pods contain considerable amounts of active compounds such as fiber, minerals, and (poly)phenols [10]. Dietary (poly)phenols are phytochemicals naturally present in many plants such as vegetables, fruits, coffee, or tea (among others) [11,12]. These secondary metabolites are well recognized as active molecules able to exert potent antioxidant and anti-inflammatory properties [13]. Previous scientific studies carried out in vitro and in vivo have highlighted the protective action of dietary (poly)phenols in the protection against various human diseases at habitual intake levels [14,15]. Although the nutritional composition of pea pods has been widely investigated, limited data about the polyphenol fraction of pea pods are currently available. Recently, Hadrich et al. [16] investigated the total content of flavonoids and (poly)phenols in pea pods extracted with different solvents. The results showed that the pea pods possess a remarkable content of these active molecules. (Poly)phenols in pea pods samples were previously analyzed using ultra-high pressure liquid chromatography (UHPLC) coupled to high-resolution mass spectrometers (HRMS) without quantitative analysis. Among analytical methods, HRMS such as Q-Orbitrap, in combination with UHPLC, represent a valid choice for identifying and quantifying (poly)phenols in foods and vegetal matrices [17,18].

As reported by scientific evidence, bioactive molecules such as (poly)phenols display a wide range of pharmacological activities, becoming suitable ingredients for nutraceutical applications [19,20,21]. To properly exert their beneficial bioactivities on human health, (poly)phenols need to achieve the target tissue [22]. According to scientific evidence, increasing the bioaccessibility of dietary (poly)phenols represents one of the key factors to maximize the bioactive properties of these molecules [23,24]. Previous studies have evidenced that both gut microbial and human digestive enzymes appear to be crucial in polyphenol adsorption [25]. Furthermore, during gastrointestinal (GI) digestion, a wide range of factors affect the bioaccessibility of the (poly)phenols such as pH and temperature variations, bacterial microflora, or digestive enzymes, which may also alter their health benefits [26]. Potential useful techniques have been developed to address this limitation [27]. As indicated by the recent literature, using nutraceutical forms to improve polyphenol bioaccessibility is recognized as a suitable strategy to protect the chemical properties of (poly)phenols that may be compromised during GI digestion [27,28].

Hence, this work aimed to obtain a comprehensive investigation of the polyphenolic fraction of the pea pods water-based extracts using a UHPLC-Q-Orbitrap HRMS methodology. Moreover, differences in the antioxidant capacity and polyphenol bioaccessibility of two nutraceutical formulations containing the water extract of the pea pods during a simulated GI digestion were also evaluated in order to select the most effective delivery mechanism capable of preserving antioxidant molecules during the GI process. 

## 2. Materials and Methods

### 2.1. Sampling

A total of ten different samples of green pea (*Pisum sativum* L.) were grown in different fields located in Campania, South of Italy, and provided by several farmers. All pea samples were harvested in May 2021. The pea pod was manually separated from the seeds, cut into small pieces, freeze-dried, pulverized to fine powder, and then stored.

### 2.2. Reagents and Materials

Standard (purity > 98%) flavonoids and phenolic acids including quinic acid, gallic acid, protocatechuic acid, epicatechin, 5-caffeoylquinic acid, catechin, *p*-cumaric acid, apigenin-7-*O*-glucoside, ferulic acid, naringin, rutin, quercetin 3 galattoside, hesperidin, rosamarinic acid, kaempferol 3 glucoside, ellagic acid, diosmin, genistein, isorhamnetin 3 rutinoside, myricetin, daidzein, quercetin, naringenin, gallic acid, luteolin, and apigenin were provided from Sigma-Aldrich (Milan, Italy). For the antioxidant experiments, 2,2′-azino-bis-3-ethylbenzthiazoline-6-sulphonic acid (ABTS), 6-hydroxy-2,5,7,8-tetramethylchromane-2-carboxylic acid (Trolox), gallic acid, potassium persulphate, 2,3,5-triphenyltetrazolio chloride (TPTZ), 1,1-diphenyl-2-picrylhydrazyl (DPPH), anhydrous ferric chloride, sodium acetate, and hydrochloric acid were provided from Sigma-Aldrich. Standards and enzymes used to simulate GI process were: pepsin, bile salt, α-amylase, pancreatin, bacterial protease from Streptomyces griseus (Pronase E), Viscozyme L and pancreatin were provided from Sigma-Aldrich. Moreover, Megazyme^®^ phytic acid assay kit, ammonium molybdate, potassium persulfate, calcium chloride, sodium carbonate, hydrochloric acid, acetate buffer, ferric chloride (FeCl_3_) and sodium chloride were provided from Sigma-Aldrich.

### 2.3. (Poly)phenols Extraction 

Water-based extract from a pea pod was obtained following a procedure previously described [29]. In short, 10 g of pulverized samples were added to 200 mL of hot water (80 °C). Afterward, the samples were stirred at 350× *g* for 10 min at 80 °C using a shaking water bath (KS130 Basic IKA, Argo Lab, Milan, Italy) and then centrifuged at 5000× *g* for 5 min. The supernatants were collected, and the residue pellets were re-extracted using the same methodology. Then, the supernatants were pooled and freeze-dried. Finally, NARC (gelatin capsules) and ARC (hydroxypropyl methylcellulose) capsules (pharmaceutical grade) containing 0.500 g of pea pod polyphenolic extract were employed. Control capsules (CT), CT-NARC and CT-ARC were prepared using 500 mg of cellulose in substitution of the pea pod extract.

### 2.4. Phytic Acid Concentration

The phytic acid concentration of the pea pod water-based extract was evaluated following the protocol described by McKie [30] using the Megazyme® phytic acid assay kit (Dublin, Ireland) [31]. In brief, phytic acid extraction was performed using 0.66 M HCl solution. Afterward, different enzymatic reactions were carried out to release inorganic phosphorus, and then, the solution was reacted with ammonium molybdate. Phytic acid content was determined by evaluating the absorbance at 655 nm of the final mixture containing molybdenum blue. The data were expressed as mg of phytic acid per 100 g of sample.

### 2.5. UHPLC and Orbitrap HRMS Analysis

Chromatographic separation of the studied polyphenol compounds was performed through an UHPLC system prepared with an autosampler device, a degassing system and, a quaternary UHPLC pump [32]. Polyphenol separation was achieved using a Kinetex column F5 (50 mm × 2.1 mm, 1.7 µm particle size, Phenomenex, Torrance, CA, USA). The mobile phase, H_2_O (A) and MeOH (B) prepared at 0.1% of FA *v/v*, was pumped at a flow rate of 0.5 mL/min, whereas the injection volume was 5 µL. The gradient elution was programmed as follows: initial 100% A for 1 min, and then decreased to 20% A in 2 min. The gradient decreased again to 0% A in 3 min. After that, the gradient switched back returning to the initial 100% A in 2 min. For column re-equilibration, it was held for another 2 min.

A Q-Exactive Orbitrap mass spectrometer set in negative mode was coupled to the UHPLC system. Full ion MS data were acquired at a resolving power of 35,000 full width at half maximum, automatic gain control target of 1 × 10^6^, maximum injection time of 200 ms, scan range 80–1200 *m/z*, microscans 1, spray voltage of 3.5 kV, and capillary temperature set at 320 °C. Detection was achieved considering the exact mass with a mass error < 5 ppm. Data analysis was performed using Xcalibur software 3.1.66.19.

### 2.6. In Vitro GI Digestion

Simulated in vitro GI digestion was carried out on the pea pod water-based extract encapsulated in NARC and ARC following the method previously reported by the INFOGEST network [33]. The simulated solutions (salivary fluid, SSF; gastric fluid, SGF; and intestinal fluid, SIF) were obtained following the proportion salts described in a previous scientific work [34].

In short, pea pods water-based extracts encapsulated in NARC and ARC was mixed with 975 µL of water, 3.5 mL of SSF, 25 µL of 0.3 M calcium chloride, and 0.5 mL of α-amylase solution. The pH of the mixture was adjusted to 7 with NaOH 1 M before incubation at 37 °C for 30 min. Subsequently, the gastric conditions were simulated by adding to the mixture 5 µL of 0.3 M calcium chloride, 7.5 mL of SGF, 1.6 mL pepsin solution, and 0.695 mL of H_2_O. The pH of the mixture was adjusted to 3 with HCl 1 M before incubation at 37 °C for 2 h. Afterward, to simulate the intestinal phase, 40 µL of 0.3 M calcium chloride, 5 mL pancreatin solution, 1.3 mL of water, 2.5 mL bile salt solution, and 11 mL of SIF were added to the mixture. Moreover, before the incubation at 37 °C for 2 h, 1 M NaOH was used to adjust the pH of the solution to 7.

Soluble fractions obtained at each in vitro digestion phase were collected in order to monitor the variation in polyphenol bioaccessibility and antioxidant activity. After centrifugation at 5000× *g* for 10 min, the remaining pellet was treated according to the protocol described elsewhere in order to simulate the colonic phase [13]. In brief, 5 mL of Pronase solution (1 mg/mL) were added to the sample. Then, the mixture was incubated at 37 °C for 60 min (pH 8). Moreover, Viscozyme L (150 µL) and 5 mL of H_2_O were added to the sample. Finally, the mixture was incubated at 37 °C for 16 h (pH 4) before centrifugation at 5000× *g* for 10 min. The supernatants were recovered and lyophilized.

### 2.7. Antioxidant Capacity

The antioxidant capacity of the pea pod water-based extract encapsulated in NARC and ARC formulations subjected to GI digestion was assessed spectrophotometrically by using three different assays namely FRAP, DPPH, and ABTS. The obtained data were expressed as mmol Trolox equivalents per kg of sample.

#### 2.7.1. FRAP Assay

The FRAP test was performed based on the methodology previously reported by Benzie et al. [35]. Briefly, acetate buffer (0.3 M, pH 3.6), FeCl_3_ solution (20 mM), and TPTZ solution (10 mM) in 0.04 M HCl were mixed with a ratio of 10:1:1 (*v/v*). Afterward, 2.85 mL of FRAP working solution (WS) were added to 150 µL of the diluted sample. The absorbance of the mixture was monitored after 4 min at 593 nm.

#### 2.7.2. DPPH Assay

The DPPH method was performed through the protocol described by Brand-Williams et al. [36]. In short, DPPH WS was prepared suspending 5 mg of DPPH standard in MeOH until reaching an absorbance value of 0.90 (at 517 nm). Then, 0.200 mL of the sample were added to 1 mL of DPPH WS. The decreased absorbance was monitored after 10 min of incubation.

#### 2.7.3. ABTS Assay

The ABTS assay was performed according to the protocol previously reported by Dini et al. [37]. In short, ABTS WS was prepared suspending 19.2 mg of ABTS standard in 5 mL of H_2_O and 88 µL of potassium persulfate (2.5 mM). After that, the mixture was diluted with EtOH to reach an absorbance value of 0.70 (at 744 nm). Then, 0.1 mL of the sample was added to 1 mL of ABTS WS. The decreased absorbance was monitored after 3 min of incubation (at 744 nm).

### 2.8. Determination of Total Phenolic Content

The Folin–Ciocalteu test was carried out to evaluate the TPC value of the assayed samples in accordance with the procedure reported by Izzo et al. [38]. In brief, 0.500 mL of H_2_O, 0.125 mL of the Folin-Ciocolteu reagent, and 0.125 mL of diluted sample were mixed and incubated for 6 min. After that, 1 mL of water and 1.25 mL of sodium carbonate solution (7.5% *w/v*) were added to the mixture. The absorbance was recorded after 60 min of incubation at 760 nm. The results were expressed as milligrams of gallic acid equivalent per gram of dry extract mg (GAE)/g.

### 2.9. Data Analysis

Tukey’s test was performed to measure differences between ARC and NARC samples. The *p*-values < 0.05 were considered as significant. The statistical software Stata 12.0 was used to perform data analysis

## 3. Results

### 3.1. Phytic Acid Concentration

The phytic acid concentration in the pea pods water-based extracts was measured by enzymatic assay. The assayed samples showed a phytic acid concentration in a range from 51.6 to 65.3 mg/100 g of samples (average content 59.6 mg/100 g).

### 3.2. Identification of Active Compounds in the Pea Pod Water-Based Extracts

A total of 24 polyphenolic compounds including flavonoids (*n* = 17) and phenolic acids (*n* = 7) were investigated in the pea pod extract by UHPLC-Q-Orbitrap HRMS analysis. The results showed that the studied analytes were optimally separated by using the UHPLC gradient system in a nine-minute run. Appendix A shows the chromatographic and mass parameters. All results were recorded in full-scan HRMS, which allowed retrospective data analysis. The structural isomers rutin and hesperidin (*m/z* 609.14611), apigenin-7-*O*-glucoside and genistein (*m/z* 269.04554) were identified by comparing the retention times (RTs) of the standards with those of the peaks. Total Ion Chromatogram (TIC) obtained through UHPLC-Q-Orbitrap HRMS analysis and the plots of twenty-one representative extracted ion chromatograms are presented in Appendix A and Appendix A (Appendix A). 

### 3.3. Quantification of Active Compounds in the Pea Pod Water-Based Extracts

Quantification of the main phenolic acids and flavonoids in pea pods water-based extracts was performed throughout a UHPLC-Q-Orbitrap HRMS method.

Calibration curves prepared in triplicate at twelve concentration levels were used in the quantitative analysis of all studied compounds. All regression coefficients obtained were >0.990. The total amount of phenolic acids present in the pea pods extracts was quantified at a concentration up to 73.16 mg/100 g, as shown in Table 1. Phenolic acids represented about 40.9% of total polyphenolic compounds found in the assayed samples. Moreover, 5-caffeoylquinic acid was the most commonly detected phenolic acid in pea pod extract, with a mean value of 59.87 mg/100 g. 

Furthermore, some important flavonoids including flavones, flavanols, flavanones, flavonols, and isoflavone were quantified in tested extracts. Flavanols, mainly represented by catechin and epicatechin, were demonstrated to be the most relevant flavonoids quantified at sum concentration of 46.33 mg/100 g, representing about 25.9% of total polyphenolic compounds present in pea pod extracts. As far as flavanones were concerned, hesperidin represented 11.9% of total polyphenolic compounds and was quantified at a mean value of 19.94 mg/100 g. Concerning the concentration of flavonols found in the assayed samples, rutin with a mean value of 14.63 mg/100 g, was the predominant analyte identified. However, *p*-coumaric acid, ferulic acid, protocatechuic acid, apigenin-7-*O*-glucoside, naringin, isorhamnetin 3-rutinoside, and daidzein were not detected in the assayed samples.

### 3.4. Bioaccessibility of Water-Based Extract of Pea Pod in NARC and ARC

Simulated in vitro GI digestion was performed on the pea pods water-based extracts encapsulated in NARC and ARC formulations in order to obtain valuable data on the ability of these formulations to protect (poly)phenols during the GI process. The TPC data were recorded after all phases of the in vitro GI digestion, as shown in Table 2. 

Compared with the TPC value found in the pea pod extract, the data highlighted a significant decrease (*p*-value < 0.05) in the TPC after all stages of GI digestion in both NARC and ARC formulations. Oral bioaccessibility was 0 mg GAE/g for both NARC and ARC formulations, while during the gastric step, the TPC value was 0 mg GAE/g only for the extract contained in ARC formulation.

Regarding the comparison between the NARC and ARC formulations, the data showed that the ARC capsules possessed significantly higher TPC values (*p*-value ≤ 0.05) after duodenal and colonic stages (Pronase E plus Viscozyme L phase) than the NARC capsules. Furthermore, the highest values of TPC were displayed after the colonic phase for both NARC and ARC formulations. The TPCs recorded after the total colon stage (Pronase E plus Viscozyme L phase) were quantified at a mean value of 5.47 and 3.73 mg GAE/g in ARC and NARC formulations, respectively. Concerning the control samples, not detected concentrations were recorded for both CT-NARC and CT-ARC during all stages of GI digestion (Appendix A, Appendix A). Moreover, the not-encapsulated extracts were also subjected to GI digestion and the results are presented in Appendix A (Appendix A).

### 3.5. Antioxidant Capacity of Pea Pod Water-Based Extracts Encapsulated in the NARC and ARC Formulations

The antioxidant capacity of the water-based extract from pea pod encapsulated in NARC and ARC was evaluated by using three different assays namely FRAP, DPPH, and ABTS during each phase of the GI digestion, in order to select the most effective delivery mechanism capable of preserving antioxidant molecules during the GI process. Table 3 shows the data expressed as mmol Trolox equivalents per kg of the sample found in each phase of the GI digestion.

In all studied experiments, compared to the non-digested samples, the antioxidant activity of both NARC and ARC samples decreased significantly (*p*-value ≤ 0.05) following all stages of GI digestion. In the colonic stage phase, however, both NARC and ARC highlighted the strongest antioxidant capacity recorded during the simulated GI digestion. Sum of Viscozyme phase plus Pronase E phase data was considered as total colonic stage value. Furthermore, in all performed spectrophotometric tests, the highest antioxidant capacity recorded during simulated GI digestion was revealed by ARC samples compared to NARC samples in both duodenal and colon stages (*p*-value ≤ 0.05). On the other hand, the antioxidant capacity of not-encapsulated extract was evaluated during simulated GI digestion and the results are shown in Appendix A (Appendix A). The TPC values recorded along the GI digestion were correlated with the FRAP, DPPH, and ABTS data (Appendix A). Regarding the control samples, Appendix A displays the data recorded for CT-NARC and CT-ARC during the simulated GI digestion.

## 4. Discussion

Overall, the findings of this study suggest that pea pod material could be an alternative source of polyphenolic compounds, including benzoic and cinnamic acids, as well as some other important phenolic compounds. The total concentration of phenolic acids displayed by water-based extracts from pea pod was 73.16 mg/100 g of the sample. In addition, flavonoids such as flavones, flavanols, flavanones, flavonols, and isoflavones were investigated in the analyzed extracts (sum average of 105.63 mg/100 g). Polyphenolic compounds have been barely studied in pea pod material [10,16,41]. To the best of our knowledge, the present work is the first report that provides a quantitative analysis of (poly)phenols of water-based extract from pea pod using an UHPLC-Q-Orbitrap HRMS method.

The aim of this study was to provide relevant data about bioactive compounds present in pea pod material, namely (poly)phenols, which may be successfully employed in nutraceutical formulations. A simple water-based extraction was performed in order to achieve a food-grade ingredient from pea pod material. Moreover, low levels of phytic acids were found in the obtained extracts. Although previous findings have reported harmful effects related to the consumption of phytic acids [39], some evidence also reports benefits for human health from the inclusion of low levels of phytic acids in the diet [40].

Recently, Guo et al. [42] studied the polyphenolic profile of pea pod by UHPLC coupled to MS technologies and a total of 31 (poly)phenols were tentatively identified including 24 flavonoids, 4 phenolic acids, and 3 other phenolics. Moreover, in another study, Guo et al., [43] performed a quantitative analysis of phenolics in pea pods extracts, reporting higher amounts of the phenolic compounds than the current study. The predominant phenolic substances found were quercetin, kaempferol trihexanside, and catechin reaching 283.6, 148.2, and 134.0 mg quercetin/100 g of the extract, respectively. The assayed extracts were obtained with a non-food-grade mixture such as methanol-water 80:20 (*v/v*), which may explain the different findings observed. Furthermore, in vivo studies reported that pea pods extracts exhibited hypocholesterolemic effects and antioxidant activities in the plasma and different tissues of rats, which could be due to the phenolic compounds found in the investigated extracts [42,44]. In addition to (poly)phenols, Belghith-Fendri et al., [45] reported that the polysaccharides extracted from pea pods possessed an important antibacterial and antioxidant activity and could be employed as an ingredient in food and cosmetic preparations. 

Although polyphenol regular intake has been linked with a broad number of biological actions involved in maintaining health status and preventing a wide range of degenerative pathologies [46,47], it is essential to take into consideration that these active molecules are highly sensitive to various factors of the GI digestion [24], which affect the chemical structure of these plant-based compounds as well as their bioactive properties [48]. The bioaccessibility of (poly)phenols has been recognized as one of the most important factors influencing the positive effects of these active ingredients [49]. Due to the susceptibility of (poly)phenols to the digestive conditions, the use of capsules capable of delivering bioactive compounds to target tissues could represent a useful strategy to be used in the development of dietary supplements and nutraceutical products [50]. 

Therefore, the main goal of this research was to obtain an overview of the changes in antioxidant activity and bioaccessibility of (poly)phenols recorded throughout simulated GI digestion of two different nutraceutical formulations such as NARC and ARC containing pea pods water-based extracts. As the INFOGEST protocol does not cover the activity of the intestinal microbiota, Pronase E and Viscozyme L have been employed to replicate the enzymatic activity carried out in the large human intestine [13]. Pronase E is made up of bacterial proteases, while Viscozyme L is an enzyme complex of different carbohydrases, including β-glucanase, hemicellulase, arabanase, xylanase, and cellulase. Although it is recognized that protocols that use fecal inoculum to study colon digestion are the most accurate in mimicking microbiota activity [51], a growing number of studies have proposed the use of Pronase E and Viscozyme L as a valid alternative to reproduce the intestinal microbiota activity [52,53,54,55].

Overall, the data clearly revealed strong correlations among FRAP, DPPH, and ABTS tests and TPC values recorded during simulated GI digestion, underlining that the tested methodologies offer accurate information on the active molecules released by the NARC and ARC formulations following simulated GI digestion. Our findings showed that in both the oral and gastric phases, the bioaccessibility of the (poly)phenols of the pea pod extract contained in the ARC formulation was 0 mg GAE/g, highlighting that the capsules were unaffected during both phases of digestion. As for the NARC formulation, only the bioaccessibility of polyphenol recorded during the oral phase was 0 mg GAE/g. As expected, there was a release of active compounds contained in them during the gastric phase. Interestingly, the results showed that both the NARC and ARC formulations showed higher TPC content and antioxidant activity in the duodenum and colonic stages compared to the not-encapsulated extracts subjected to GI digestion, showing the ability of the tested capsules to protect active compounds during the simulated GI digestion. Nevertheless, scientific evidence suggested that some important flavanones and flavanols, found in the analyzed samples, were stable under gastrointestinal conditions [56,57]. Moreover, when compared to NARC samples, the pea pods water-based extracts contained in ARC formulation highlighted a significantly higher antioxidant activity and TPC value in both the duodenum and colonic stages. These findings are in agreement with a previous study that reports that water-based extracts from fennel waste in the ARC formulation showed high polyphenol bioaccessibility in the colonic stage after simulated GI digestion [29]. Moreover, Izzo et al. [58] also demonstrated that red cabbage extract contained in ARC formulations showed higher colon bioaccessibility than the not-encapsulated extract. In this line, Amrani-Allalou et al. [59] reported a higher TPC value after the GI process in plant extracts encapsulated in ARC compared to the same extracts tested without capsules. 

These data suggest that ARC may be able to protect active molecules from the gastric environment, preserving their chemical structure and bioactivity. Hence, ARC may be a practical strategy for delivering active compounds to specific tissues where they can exert their health-promoting bioactivities.

## 5. Conclusions

In conclusion, the present study reported the polyphenolic profile including flavonoids (*n* = 17) and phenolic acids (*n* = 7) of pea pods extracts throughout a UHPLC-Q-Orbitrap HRMS analysis. The most commonly detected polyphenolic compounds were mainly represented by 5-caffeoylquinic acid, epicatechin, hesperidin, and catechin (average content: 59.87, 29.46, 19.94, 16.87 mg/100 g, respectively). Furthermore, the results of this study suggested that the ARC formulations were able to preserve the active compounds along the simulated GI process, highlighting a higher TPC value and antioxidant capacity than the NARC formulations. Overall, the pea pods water-based extracts could be utilized as an innovative ingredient rich in polyphenolic compounds and the use of ARC can be considered a useful strategy for delivering active compounds to specific tissues.

However, further in-depth studies are needed to determine the many biotransformations involving polyphenolic compounds during in vivo GI digestion in order to estimate the efficacy of the obtained ingredient for future nutraceutical applications.

## Figures and Tables

**Table 1 antioxidants-11-00105-t001:** Phenolic acids and flavonoids content in the pea pods water-based extracts.

Compounds	Averange (mg/100 g)	±SD
PHENOLIC ACIDS		
Cinnamic acid		
Quinic acid	7.20	0.91
5-caffeoylquinic acid	59.87	0.73
*p*-Coumaric acid	N.D.	
Ferulic acid	N.D.	
Rosamarinic acid	0.78	0.31
SUM	67.85	0.65
Benzoic Acid		
Gallic acid	5.30	0.69
Protocatechuic acid	N.D.	
SUM	5.30	0.69
FLAVONOIDS		
Flavones		
Luteolin	0.42	0.12
Apigenin-7-*O*-glucoside	N.D.	
Apigenin	0.12	0.01
Diosmin	<LOD	
Kaemferol 3-glucoside	3.50	0.16
SUM	3.62	0.10
Flavanols		
Catechin	16.87	0.10
Epicatechin	29.46	0.41
SUM	46.33	0.25
Flavanones		
Naringenin	1.67	0.04
Naringin	N.D.	
Hesperidin	19.94	3.62
SUM	21.61	1.83
Flavonols		
Isorhamnetin 3-rutinoside	N.D.	
Quercetin	2.70	0.09
Quercetin 3-galattoside	0.73	0.00
Rutin	14.63	1.47
SUM	18.06	0.52
Isoflavone		
Genistein	7.41	0.05
Myricetin	8.59	0.09
Daidzein	N.D.	
SUM	16.00	0.07
TOTAL (POLY)PHENOLS	178.79	0.59

N.D., not detected.

**Table 2 antioxidants-11-00105-t002:** Bioaccessibility of polyphenolic compounds in pea pods water-based extracts encapsulated in the NARC and ARC formulation.

Samples	TPC mg GAE/g ± SD	
Pea pod extract	8.22 ± 0.31	
	NARC	ARC
Digestion Stage		
Oral stage	N.D.	N.D.
Gastric stage	1.12 ± 0.03	N.D.
Duodenal stage	1.84 ± 0.08 *	2.01 ± 0.03 *
Pronase E	1.95 ± 0.07 *	3.39 ± 0.06 *
Viscozyme L	1.78 ± 0.08 *	2.08 ± 0.07 *
Total colonic stage	3.73 ± 0.08 *	5.47 ± 0.07 *

Differences between groups were statistically analyzed with Tukey’s test; * *p*-value ≤ 0.05 was considered significant. Abbreviations: mg GAE/g: milligrams of gallic acid equivalent per gram of dry extract; N.D., not detected.

**Table 3 antioxidants-11-00105-t003:** Antioxidant capacity of NARC and ARC formulations evaluated by FRAP, DPPH, and ABTS assays during simulated GI digestion.

	DPPH mmol/kg ± SD		ABTS mmol/kg ± SD		FRAP mmol/kg ± SD	
Not Digested	13.7 ± 1.1		18.3 ± 1.2		12.3 ± 1.1	
	NARC	ARC	NARC	ARC	NARC	ARC
Digestion stage						
Oral stage	N.D.	N.D.	N.D.	N.D.	N.D.	N.D.
Gastric stage	1.6 ± 0.1	N.D.	2.0 ± 0.3	N.D.	0.9 ± 0.1	N.D.
Duodenal stage	2.1 ± 0.5	2.6 ± 0.4	3.2 ± 0.2	4.0 ± 0.5	1.4 ± 0.1 *	2.1 ± 0.2 *
Pronase E stage	2.3 ± 0.3 *	4.5 ± 0.5 *	4.6 ± 0.4 *	5.9 ± 0.5 *	3.4 ± 0.3 *	4.6 ± 0.4 *
Viscozyme L stage	1.9 ± 0.2	2.1 ± 0.5	2.8 ± 0.3 *	3.6 ± 0.5 *	2.4 ± 0.2 *	3.9 ± 0.3 *
Total colonic stage	4.2 ± 0.3 *	6.6 ± 0.5 *	7.4 ± 0.4 *	9.5 ± 0.5 *	5.8 ± 0.3 *	8.5 ± 0.4 *

Differences between groups were statistically analyzed with Tukey’s test; * *p*-value ≤ 0.05 was considered significant. Abbreviations: N.D., not detected.

## Data Availability

Data is contained within the article or Appendix A.

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
