# Peer review of "Chemical Composition of Green Pea (Pisum sativum L.) Pods Extracts and Their Potential Exploitation as Ingredients in Nutraceutical Formulations"

_antioxidants, 2021, doi:10.3390/antiox11010105_

Round 1
Reviewer 1 Report
357-9 The most commonly detected polyphenolic compounds were 357 mainly represented by chlorogenic acid, epicatechin, hesperidin, and catechin (average 358 content: 59.87, 29.46, 19.94, 16.87 mg/100 g, respectively). Discuss how this levels of polyphenolic compounds are to desired concentration of these compounds in the eatable plants.
The paper is interesting. There are in the literature papers on green pea hulls, pods or peels polyphenol composition and green pea polyphenol composition. Different names for pea hulls is used, so sometimes it is difficult to see the similarities between different publications. Pea pods water-based extracts were analyzed using through a UHPLC‐Q‐Orbitrap HRMS. In this approach the paper contains novelty. Polyphenols in pea pods samples were previously analyzed using ultra high pressure liquid chromatography (UHPLC) coupled to high resolution mass spectrometers (HRMS) without quantitative analysis. In the present studies quantification of the main phenolic acids and flavonoids in pea pods water based extracts was performed throughout a UHPLCQ Orbitrap HRMS method.
The paper is written in good English. The results are interesting and many analytical methods were used.
Some important papers are missing and should be added to the text.
Guo et al. (2019) Phenolics of Green Pea (Pisum sativum L.) Hulls, Their Plasma and Urinary Metabolites, Bioavailability, and in Vivo Antioxidant Activities in a Rat Model. JOURNAL OF AGRICULTURAL AND FOOD CHEMISTRY, 67(43), 1955-11968.
DOI 10.1021/acs.jafc.9b04501
Increased processing of pulses generates large volumes of hulls, which are known as an excellent source of phenolic antioxidants. However, the bioavailability and in vivo activity of these phenolics are rarely reported. This research was therefore carried out to study the absorption, metabolism, and in vivo antioxidant activities of green pea hull (GPH) phenolics using ultrahigh-pressure liquid chromatography with a linear ion trap-high-resolution Orbitrap mass spectrometry and an oxidative stress rat model. A total of 31 phenolics, including 4 phenolic acids, 24 flavonoids, and 3 other phenolics, were tentatively identified. Ten of these phenolics and 49 metabolites were found in the plasma and urine of rats, which helped to explain the favorable changes by GPH phenolics in key antioxidant enzymes (superoxide dismutase, glutathione peroxidase, and glutathione) and indicators (total antioxidant capacity, malondialdehyde) in the plasma and different tissues of rats. This is the first comprehensive report on dry pea hull phenolics and their bioavailability, metabolic profiles, and mechanisms of in vivo antioxidant activities.
In another paper Guo et al. (2021) investigated Green Pea (Pisum sativum L.) Hull Polyphenol Extracts Ameliorate DSS-Induced Colitis through Keap1/Nrf2 Pathway and Gut Microbiota Modulation. FOODS, 10(11), 2765.
DOI 10.3390/foods10112765
In this study, UHPLC-LTQ-OrbiTrap-MS (Ultra performance liquid chromatography-linear ion trap orbitrap tandem mass spectrometry) technique was used to quantify polyphenols, and DSS (sodium dextran sulfate)-induced colitis mouse model was established to explore the effect of GPH extracts on colitis. The results showed that quercetin and its derivatives, kaempferol trihexanside and catechin and its derivatives were the main phenolic substances in the extract, reaching 2836.57, 1482.00 and 1339.91 mu g quercetin/g GPH extract, respectively; GPH extracts can improved inflammatory status, repaired colonic function, regulated inflammatory factors, and restored oxidative balance in mice. Further, GPH extracts can activate Keap1-Nrf2-ARE signaling pathway, regulate downstream antioxidant protease and gut microbiota by increasing F/B value and promoting the growth of Lactobacillaceae and Lachnospiraceae, and improve the level of SCFAs (short-chain fatty acids) to relieve DSS-induced colitis in mice. Therefore, GPH may be a promising dietary resource for the treatment of ulcerative colitis.
Good paper, can be published after minor revision
Author Response
Manuscript ID: antioxidants-1513185
Type of manuscript: Article
Title: Chemical Composition of Green Pea (Pisum sativum L.) Pods Extracts and their Potential Exploitation as Ingredients in Nutraceutical Formulations
Reviewer 1
1) The most commonly detected polyphenolic compounds were 357 mainly represented by chlorogenic acid, epicatechin, hesperidin, and catechin (average content: 59.87, 29.46, 19.94, 16.87 mg/100 g, respectively). Discuss how this levels of polyphenolic compounds are to desired concentration of these compounds in the eatable plants.
1) The authors thank Reviewer 1 for valuable advice and added the missing information in the discussion section as “Moreover, in another study, Guo et al., [41] performed a quantitative analysis of phenolics in pea pods extracts, reporting higher amounts of the phenolic compounds than the current study. The predominant phenolic substances found were quercetin, kaempferol trihexanside, and catechin reaching 283.6, 148.2, and 134.0 mg quercetin/100 g of extract, respectively. The assayed extracts were obtained with a not food-grade mixture such as methanol-water 80:20 (v:v), which may explain the different findings observed”
The paper is interesting. There are in the literature papers on green pea hulls, pods or peels polyphenol composition and green pea polyphenol composition. Different names for pea hulls is used, so sometimes it is difficult to see the similarities between different publications. Pea pods water-based extracts were analyzed using through a UHPLC‐Q‐Orbitrap HRMS. In this approach the paper contains novelty. Polyphenols in pea pods samples were previously analyzed using ultra high pressure liquid chromatography (UHPLC) coupled to high resolution mass spectrometers (HRMS) without quantitative analysis. In the present studies quantification of the main phenolic acids and flavonoids in pea pods water based extracts was performed throughout a UHPLCQ Orbitrap HRMS method. The paper is written in good English.
2) The results are interesting and many analytical methods were used. Some important papers are missing and should be added to the text. Guo et al. (2019) Phenolics of Green Pea (Pisum sativum L.) Hulls, Their Plasma and Urinary Metabolites, Bioavailability, and in Vivo Antioxidant Activities in a Rat Model. JOURNAL OF AGRICULTURAL AND FOOD CHEMISTRY, 67(43), 1955-11968. DOI 10.1021/acs.jafc.9b04501 Increased processing of pulses generates large volumes of hulls, which are known as an excellent source of phenolic antioxidants. However, the bioavailability and in vivo activity of these phenolics are rarely reported. This research was therefore carried out to study the absorption, metabolism, and in vivo antioxidant activities of green pea hull (GPH) phenolics using ultrahigh-pressure liquid chromatography with a linear ion trap-high-resolution Orbitrap mass spectrometry and an oxidative stress rat model. A total of 31 phenolics, including 4 phenolic acids, 24 flavonoids, and 3 other phenolics, were tentatively identified. Ten of these phenolics and 49 metabolites were found in the plasma and urine of rats, which helped to explain the favorable changes by GPH phenolics in key antioxidant enzymes (superoxide dismutase, glutathione peroxidase, and glutathione) and indicators (total antioxidant capacity, malondialdehyde) in the plasma and different tissues of rats. This is the first comprehensive report on dry pea hull phenolics and their bioavailability, metabolic profiles, and mechanisms of in vivo antioxidant activities.
- As suggested by Reviewer 1, the authors corrected the citation, which should have referred to the article reported by Reviewer 1.
3) In another paper Guo et al. (2021) investigated Green Pea (Pisum sativum L.) Hull Polyphenol Extracts Ameliorate DSS-Induced Colitis through Keap1/Nrf2 Pathway and Gut Microbiota Modulation. FOODS, 10(11), 2765. DOI 10.3390/foods10112765. In this study, UHPLC-LTQ-OrbiTrap-MS (Ultra performance liquid chromatography-linear ion trap orbitrap tandem mass spectrometry) technique was used to quantify polyphenols, and DSS (sodium dextran sulfate)-induced colitis mouse model was established to explore the effect of GPH extracts on colitis. The results showed that quercetin and its derivatives, kaempferol trihexanside and catechin and its derivatives were the main phenolic substances in the extract, reaching 2836.57, 1482.00 and 1339.91 mu g quercetin/g GPH extract, respectively; GPH extracts can improved inflammatory status, repaired colonic function, regulated inflammatory factors, and restored oxidative balance in mice. Further, GPH extracts can activate Keap1-Nrf2-ARE signaling pathway, regulate downstream antioxidant protease and gut microbiota by increasing F/B value and promoting the growth of Lactobacillaceae and Lachnospiraceae, and improve the level of SCFAs (short-chain fatty acids) to relieve DSS-induced colitis in mice. Therefore, GPH may be a promising dietary resource for the treatment of ulcerative colitis.
- As suggested by Reviewer 1, the authors added the missing information as ” Moreover, in another study, Guo et al., [43] performed a quantitative analysis of phenolics in pea pods extracts, reporting higher amounts of the phenolic compounds than the cur-rent study. The predominant phenolic substances found were quercetin, kaempferol tri-hexanside, and catechin reaching 283.6, 148.2, and 134.0 mg quercetin/100 g of extract, respectively. The assayed extracts were obtained with a not food-grade mixture such as methanol-water 80:20 (v:v), which may explain the different findings observed. Further-more, in vivo studies reported that pea pods extracts exhibited hypocholesterolemic effects and antioxidant activities in the plasma and different tissues of rats, which could be due to the phenolic compounds found in the investigated extracts [42, 44]. In addition to (poly)phenols, Belghith-Fendri et al., [45] reported that the polysaccharides extracted from pea pod possessed an important antibacterial and antioxidant activity and could be em-ployed as an ingredient in food and cosmetic preparations..”
Good paper, can be published after minor revision
The authors thank Reviewer 1 for evaluating the manuscript.

Reviewer 2 Report
The manuscript titled as " Chemical Composition of Green Pea (Pisum sativum L.) Pods Extracts and their Potential Exploitation as Ingredients in Nutraceutical Formulations " is worth publishing due to its well organization, adequate level and interesting topic of discussion. However, there are some details that need to be explanation:
1) Many works describe the problematic nomenclature of isomers belonging to the chlorogenic acid family. The authors of the article "UNREMITTING PROBLEMS WITH CHLOROGENIC ACID NOMENCLATURE: A REVIEW" indicate an obvious and unremitting wrong nomenclature occurring in chlorogenic acid family. "These discrepancies are caused by the fact that 5-caffeoylquinic acid (5-CQA) was firstly discovered and subsequently isolated already in the middle of 19th century. However, from that time up to 1976, when IUPAC published the exact rules and defnitions describing a new system of nomenclature, the right name for the current 5-CQA was actually 3-caffeoylquinic acid. In spite of that fact, researchers and also chemicals suppliers have been still using this pre-IUPAC nomenclature".
For this reason, the work should give the detailed names of the reference substances.
2) The chromatogram of the separation of standard substances (the quality of their separation) should be included in supplementary.
3) In article, the authors reported that the data statistical analysis and Tukey's test were performed (chapter 2.9. Data Analysis). Where is significance in the results?
4) Incorrectly attached file in the "Non-published materia". The "supplementary materials" file is repeated.
5) Some compounds (for example, hesperidin, daidzein, apigenin and more) are missing in table S2. Why?
6) There is no precise explanation of the presented results (for example results in chapter 3.4 verse 251- mgGAE /g, table 1 - mg/100g). What does mean "g" (gram of dry extract or raw material) ?
7) In chapter 2.7.1-2.7.3 and 2.8 is no explanation how results were expressed (units).
In summary, the presented work, especially the methodology and research results should be improved and supplemented.
Therefore, I suggest major revision.
Author Response
Manuscript ID: antioxidants-1513185
Type of manuscript: Article
Title: Chemical Composition of Green Pea (Pisum sativum L.) Pods Extracts and their Potential Exploitation as Ingredients in Nutraceutical Formulations
Reviewer 2
The manuscript titled as " Chemical Composition of Green Pea (Pisum sativum L.) Pods Extracts and their Potential Exploitation as Ingredients in Nutraceutical Formulations " is worth publishing due to its well organization, adequate level and interesting topic of discussion. However, there are some details that need to be explanation:
1) Many works describe the problematic nomenclature of isomers belonging to the chlorogenic acid family. The authors of the article "UNREMITTING PROBLEMS WITH CHLOROGENIC ACID NOMENCLATURE: A REVIEW" indicate an obvious and unremitting wrong nomenclature occurring in chlorogenic acid family. "These discrepancies are caused by the fact that 5-caffeoylquinic acid (5-CQA) was firstly discovered and subsequently isolated already in the middle of 19th century. However, from that time up to 1976, when IUPAC published the exact rules and defnitions describing a new system of nomenclature, the right name for the current 5-CQA was actually 3-caffeoylquinic acid. In spite of that fact, researchers and also chemicals suppliers have been still using this pre-IUPAC nomenclature". For this reason, the work should give the detailed names of the reference substances.
1) - As suggested by Reviewer 2, the authors changed the name chlorogenic acid to 5-caffeoylquinic acid.
2) The chromatogram of the separation of standard substances (the quality of their separation) should be included in supplementary.
- As suggested by reviewer 2, Total Ion Chromatogram (TIC) obtained through UHPLC-Q-Orbitrap HRMS analysis and the plots of twenty-one representative extracted ion chromatograms were added in supplementary materials.
3) In article, the authors reported that the data statistical analysis and Tukey's test were performed (chapter 2.9. Data Analysis). Where is significance in the results?
- As suggested by Reviewer 2, the authors added the missing information
4) Incorrectly attached file in the "Non-published materia". The "supplementary materials" file is repeated.
- As suggested by Reviewer 2, the authors removed the published material in the "supplementary materials"
5) Some compounds (for example, hesperidin, daidzein, apigenin and more) are missing in table S2. Why?
- As suggested by Reviewer 2, the authors added the missing information
6) There is no precise explanation of the presented results (for example results in chapter 3.4 verse 251- mgGAE /g, table 1 - mg/100g). What does mean "g" (gram of dry extract or raw material) ?
- As suggested by Reviewer 2, the authors added the missing information as “Abbreviations: mg GAE/g: milligram of gallic acid equivalent per gram of dry extract”
7) In chapter 2.7.1-2.7.3 and 2.8 is no explanation how results were expressed (units).
- As suggested by Reviewer 2, the authors added the missing information in chapter 2.8, while the units used for antioxidant activity data are reported in chapter 2.7.
In summary, the presented work, especially the methodology and research results should be improved and supplemented.
Therefore, I suggest major revision.
The authors thank Reviewer 2 for evaluating the manuscript.

Reviewer 3 Report
Finding new uses for agricultural waste is one of the strategies followed in Europe contributing to a circular economy and complying with EU building industry standards. The investigation carried out by Castaldo et al. is in line with this strategy. The study focuses on the use of pea pods as an interesting element to obtain future nutraceuticals. The authors described the (poly)phenols composition of the pods using a water-based extraction process followed by the UHPLC analysis. Besides, the authors also studied the effects of an in vitro gastrointestinal digestion on the encapsulated extracts testing the antioxidant activity at the different steps of the digestion process.
After reading this manuscript I have a number of major concerns that should be addressed by the authors in case the handling editor invites for a resubmission:
- The authors only tested the effects of the in vitro gastrointestinal digestion in encapsulated extracts. What is the effect on non-encapsulated extracts? How do the authors know the level of protection of the capsules if the effects in their absences was overlooked? This approach is described in studies included in the references section (ref. 47 and 48) and is essential to demonstrate the capacity of the capsules to preserve the bioactive molecules.
- In this study, the authors considered that the GI digestion will modify the structure of the (poly)phenols that are present in the pea pods (justifying the use of the capsules). However, there are some of the bioactive molecules that are stable to the GI conditions such as flavanones (DOI:10.1021/jf901983g and DOI: 10.3390/antiox10020140) and quercetin (DOI: 10.1021/jf202279r). This should be discussed in the manuscript.
- Are there other compounds present in the water extracts that could be responsible for the antioxidant activity observed? An example could be polysaccharides, which can be extracted from pea pods and exert antioxidant activity (DOI: 10.1016/j.ijbiomac.2018.05.095).
- Is the consumption of pea pods-enriched diet related to beneficial effects in vivo (human and/or animal studies? If so, the studies should be included. It will give a background to justify the use of these agroindustry waste as nutraceuticals.
- Both phenolic compounds and flavonoids are referred as polyphenols through the manuscript. This is incorrect as phenolic compounds are not polyphenols. Please, re-write polyphenols to read (poly)phenols.
- The number of self-citations is higher than 20% for some of the authors. This must be checked by the authors.
Other comments:
Table 1. Total POLYPHENOLS is the value obtained from the total amount of phenolic compounds (which are not polyphenols) and the flavonoids present in the samples. Please correct this as indicated above.
Table 2. Please review the number 2..08 and re-write correctly
Table 3. Please be consistent with the way to write N.D.
Author Response
Manuscript ID: antioxidants-1513185
Type of manuscript: Article
Title: Chemical Composition of Green Pea (Pisum sativum L.) Pods Extracts and their Potential Exploitation as Ingredients in Nutraceutical Formulations
Reviewer 3
Finding new uses for agricultural waste is one of the strategies followed in Europe contributing to a circular economy and complying with EU building industry standards. The investigation carried out by Castaldo et al. is in line with this strategy. The study focuses on the use of pea pods as an interesting element to obtain future nutraceuticals. The authors described the (poly)phenols composition of the pods using a water-based extraction process followed by the UHPLC analysis. Besides, the authors also studied the effects of an in vitro gastrointestinal digestion on the encapsulated extracts testing the antioxidant activity at the different steps of the digestion process.
After reading this manuscript I have a number of major concerns that should be addressed by the authors in case the handling editor invites for a resubmission:
1) The authors only tested the effects of the in vitro gastrointestinal digestion in encapsulated extracts. What is the effect on non-encapsulated extracts? How do the authors know the level of protection of the capsules if the effects in their absences was overlooked? This approach is described in studies included in the references section (ref. 47 and 48) and is essential to demonstrate the capacity of the capsules to preserve the bioactive molecules.
- As suggested by Reviewer 3, the authors tested the effects of the in vitro gastrointestinal digestion on not-encapsulated extracts. The missing information was added to the manuscript and supplementary material.
2) In this study, the authors considered that the GI digestion will modify the structure of the (poly)phenols that are present in the pea pods (justifying the use of the capsules). However, there are some of the bioactive molecules that are stable to the GI conditions such as flavanones (DOI:10.1021/jf901983g and DOI: 10.3390/antiox10020140) and quercetin (DOI: 10.1021/jf202279r). This should be discussed in the manuscript.
- As suggested by Reviewer 3, the authors added the missing information as “Interestingly, the results showed that both the NARC and ARC formulations showed higher TPC content and antioxidant activity in the duodenum and colonic stages compared to the not-encapsulated extracts subjected to GI digestion, showing the ability of the tested capsules to protect active compounds during the simulated GI digestion. Nevertheless, scientific evidence suggested that some important flavanones and flavanols, found in the analyzed samples, were stable under gastrointestinal conditions [56, 57].”
3) Are there other compounds present in the water extracts that could be responsible for the antioxidant activity observed? An example could be polysaccharides, which can be extracted from pea pods and exert antioxidant activity (DOI: 10.1016/j.ijbiomac.2018.05.095).
- As suggested by Reviewer 3, the authors added the missing information as “In addition to (poly)phenols, Belghith-Fendri et al., [43] reported that the polysaccharides extracted from pea pod possessed an important antibacterial and antioxidant activity and could be employed as an ingredient in food and cosmetic preparations”.
4) Is the consumption of pea pods-enriched diet related to beneficial effects in vivo (human and/or animal studies? If so, the studies should be included. It will give a background to justify the use of these agroindustry waste as nutraceuticals.
- As suggested by Reviewer 3, the authors added the missing information as “Furthermore, in vivo studies reported that pea pods extracts exhibited hypocholesterolemic effects and antioxidant activities in the plasma and different tissues of rats, which could be due to the phenolic compounds found in the investigated extracts [40,42]”
5) Both phenolic compounds and flavonoids are referred as polyphenols through the manuscript. This is incorrect as phenolic compounds are not polyphenols. Please, re-write polyphenols to read (poly)phenols.
- As suggested by Reviewer 3, the authors re-wrote the term “polyphenols” to read (poly)phenols.
6) The number of self-citations is higher than 20% for some of the authors. This must be checked by the authors.
- As suggested by Reviewer 3, the authors have checked the rate of self-citations. Currently, it is less than 17%.
Other comments:
Table 1. Total POLYPHENOLS is the value obtained from the total amount of phenolic compounds (which are not polyphenols) and the flavonoids present in the samples. Please correct this as indicated above.
- As suggested by Reviewer 3, the authors re-wrote the term “polyphenols” to read (poly)phenols.
Table 2. Please review the number 2..08 and re-write correctly
- As suggested by Reviewer 3, the authors corrected the number “2.08”
Table 3. Please be consistent with the way to write N.D.
- As suggested by Reviewer 3, the authors corrected the inconsistencies.
The authors thank Reviewer 3 for evaluating the manuscript.

Reviewer 4 Report
The manuscript entitled "Chemical Composition of Green Pea (Pisum sativum L.) Pods Extracts and their Potential Exploitation as Ingredients in Nutraceutical Formulations" aimed to obtain a comprehensive investigation of the polyphenolic fraction of the pea pods water-based extracts using a UHPLC-Q-Orbitrap HRMS methodology. The manuscript suggest that pea pod material could be an alternative source of polyphenolic compounds, including benzoic and cinnamic acids, as well as some other important phenolic compounds. The experiments are well plan, the ideas and methods are scientific rigorous. It will attract the attention of a wide readership. In my mind, the manuscript is acceptable for publication in Antioxidants after minor revision. 1. ABSTRACT section need add some important data to support the conclusion. 2. add Keywords “Pea”. 3. Line 46-48 Generally, procedures that require a simple water-based extraction are considered as the greenest approaches to reduce the environmental impact and to recover active molecules from agro-industrial wastes [4, 5, 6].6 Tarun Belwal, Farid Chemat, Petras Rimantas Venskutonis, Giancarlo Cravotto, Durgesh Kumar Jaiswal, Indra Dutt Bhatt, Hari Prasad Devkota, Zisheng Luo. Recent advances in scaling-up of non-conventional extraction techniques: learning from successes and failures. Trends in Analytical Chemistry, 2020, 127:115895 4. Line 70-71 As reported by scientific evidence, bioactive molecules such as polyphenols display a wide range of pharmacological activities, becoming suitable ingredients for nutraceutical applications [15, 16, 17].17 Jarukitt Limwachiranon, Hao Huang, Zhenghan Shi, Li Li, Zisheng Luo. Lotus flavonoids and phenolic acids: health promotion and safe consumption dosages. Comprehensive Reviews in Food Science and Food Safety, 2018, 17: 458-471 5. line 285 suggest Table 3.in the same page.
Author Response
Manuscript ID: antioxidants-1513185
Type of manuscript: Article
Title: Chemical Composition of Green Pea (Pisum sativum L.) Pods Extracts and their Potential Exploitation as Ingredients in Nutraceutical Formulations
Reviewer 4
The manuscript entitled "Chemical Composition of Green Pea (Pisum sativum L.) Pods Extracts and their Potential Exploitation as Ingredients in Nutraceutical Formulations" aimed to obtain a comprehensive investigation of the polyphenolic fraction of the pea pods water-based extracts using a UHPLC-Q-Orbitrap HRMS methodology. The manuscript suggests that pea pod material could be an alternative source of polyphenolic compounds, including benzoic and cinnamic acids, as well as some other important phenolic compounds. The experiments are well plan, the ideas and methods are scientific rigorous. It will attract the attention of a wide readership. In my mind, the manuscript is acceptable for publication in Antioxidants after minor revision.
- ABSTRACT section need add some important data to support the conclusion.
- As suggested by Reviewer 4, the authors added the missing information
- add Keywords “Pea”.
- As suggested by Reviewer 4, the authors added “Pea” as Keywords.
- Line 46-48 Generally, procedures that require a simple water-based extraction are considered as the greenest approaches to reduce the environmental impact and to recover active molecules from agro-industrial wastes [4, 5, 6].6 Tarun Belwal, Farid Chemat, Petras Rimantas Venskutonis, Giancarlo Cravotto, Durgesh Kumar Jaiswal, Indra Dutt Bhatt, Hari Prasad Devkota, Zisheng Luo. Recent advances in scaling-up of non-conventional extraction techniques: learning from successes and failures. Trends in Analytical Chemistry, 2020, 127:115895 4.
- As suggested by Reviewer 4, the authors added the citation in the manuscript.
Line 70-71 As reported by scientific evidence, bioactive molecules such as polyphenols display a wide range of pharmacological activities, becoming suitable ingredients for nutraceutical applications [15, 16, 17].17 Jarukitt Limwachiranon, Hao Huang, Zhenghan Shi, Li Li, Zisheng Luo. Lotus flavonoids and phenolic acids: health promotion and safe consumption dosages. Comprehensive Reviews in Food Science and Food Safety, 2018, 17: 458-471 5.
- As suggested by Reviewer 4, the authors added the citation in the manuscript.
line 285 suggest Table 3 in the same page.
- The authors express their gratitude to Reviewer 4 for valuable advice, unfortunately, it is difficult to integrate Table 3 with the previous section.
The authors thank Reviewer 4 for evaluating the manuscript.

Round 2
Reviewer 3 Report
The authors have addressed all the questions. In my opinion, the manuscript should be accepted.
Only two comments regarding the supplementary material:
- Check the value of the ABTS analysis in the table S4 corresponding to the duodenal stage
- The title of the table S5 is out of place. Please, check it.
Congratulations on this nice paper.